# Aromatherapy in Palliative Care: A Single-Institute Retrospective Analysis Evaluating the Effect of Lemon Oil Pads against Nausea and Vomiting in Advanced Cancer Patients

**DOI:** 10.3390/cancers14092131

**Published:** 2022-04-24

**Authors:** Gudrun Kreye, Manuela Wasl, Andrea Dietz, Daniela Klaffel, Andrea Groselji-Strele, Katharina Eberhard, Anna Glechner

**Affiliations:** 1Karl Landsteiner University of Health Sciences, Dr. Karl-Dorrek-Straße 30, 3500 Krems, Austria; 2Department of Internal Medicine 2, Clinical Division of Palliative Medicine, University Hospital Krems, Mitterweg 10, 3500 Krems, Austria; manuela.wasl@krems.lknoe.at (M.W.); andrea.dietz@a1telekom.at (A.D.); daniela.klaffel@krems.lknoe.at (D.K.); 3Core Facility Computational Bioanalytics, Medical University Graz, Stiftingtalstraße 24, A-8010 Graz, Austria; andrea.groselj-strele@medunigraz.at (A.G.-S.); katharina.eberhard@medunigraz.at (K.E.); 4Department for Evidence-Based Medicine and Clinical Epidemiology, Danube University Krems, 3500 Krems, Austria; anna.glechner@donau-uni.ac.at

**Keywords:** aromatherapy, palliative care, antiemetic, nausea, vomiting

## Abstract

**Simple Summary:**

Palliative care aims to improve the quality of life for patients with serious illnesses by improving symptoms. Many patients with advanced diseases suffer from nausea and vomiting. In addition to evidenced-based treatments, patients with advanced diseases often seek for complementary and integrative therapies. Aromatherapy is commonly used to alleviate the various symptoms of palliative care patients and is well accepted by both patients and their caregivers. We retrospectively evaluated the efficacy of using lemon oil pads against nausea and vomiting in our palliative care unit and found a positive effect against nausea and vomiting in this patient group.

**Abstract:**

Aromatherapy is regularly used in the University Hospital Krems’s palliative care unit. In a retrospective analysis, we investigated whether there were improvements in nausea and vomiting in patients with advanced cancers over a time span of 24 months. Data collection used the medical records of patients who were institutionally approved to receive routine aroma applications for alleviating nausea and vomiting. The efficacy of using lemon oil pads was tested with one-dimensional chi-squared tests. Sixty-six patients received 222 applications of lemon oil on cotton pads; no data were available for 17 applications. The adequate relief of nausea and vomiting was reported for 149 (73%) applications, whereas no symptom control was seen for 56 (27%) applications. For the 56 applications without symptom control, first- and second-line rescue medications were successful in 53 and 3 cases, respectively. The use of aromatherapy with lemon oil pads against nausea and vomiting was feasible for 73% of all applications. All patients who did not benefit from aromatherapy had effective symptom control with a rescue medication. Large randomized prospective trials are necessary to evaluate the benefit of the use of lemon oil pads against nausea and vomiting in patients with advanced cancer.

## 1. Introduction

The increased use of complementary and integrative medicine (CIM) has been recognized globally in the general population, especially in cancer and palliative care patients [1,2]. The evidence of the efficacy of such treatments is low; nevertheless, patients and their relatives often use complementary and integrative therapies [3,4,5]. The quality of life is seriously impacted for patients with advanced diseases by symptoms such as pain, depression, dyspnea, constipation, as well as nausea and vomiting [6]. Very often, patients seek help in using CIM therapies to alleviate these symptoms [7].

Aromatherapy is a well-accepted CIM therapy not only by patients with advanced diseases, but also by their caregivers [8]. Aromatherapy is used in many cultures and societies, often together with conventional medicine. The exact mechanism of its therapeutic benefits as well as the correct dosage of aromatic oils remains unclear. In palliative care, aromatic oils can be used to alleviate various symptoms, such as anxiety, shortness of breath, nausea, feeling of tension in ascites, pain, coughing, nausea, tiredness, improved smell, and restlessness or simply to improve well-being.

A large systematic review of 22 trials including 1956 participants evaluated the current evidence of the effectiveness of aromatherapy, massage, and reflexology in palliative care patients [8]. This review found that although there are many trials investigating these therapies, the evidence of them is poor and heterogenous.

A comprehensive meta-analysis from 2012 did not identify any long-lasting benefits of aromatherapy [9].

A recent review evaluated studies published between 2011 and 2021 focusing on aromatherapy and textiles discussed textile materials as possible essential oil carriers [10]. Because textiles are applied to the skin, they seem to be a good delivery system for essential oils. However, the authors found significant gaps in the field. They proposed interprofessional studies for a full understanding of the therapeutic efficacy of essential oils.

Evidence of the effectiveness of CIM in palliative care is uncertain. The literature on nausea and vomiting related to aromatherapy in palliative care patients is sparse [7]. Nevertheless, many patients use these therapies [11]. A systematic review of five qualitative studies on palliative care, aromatherapy, reflexology, and massage revealed that patients with advanced cancers benefit from aromatherapy [12], reflexology, and massage through increased well-being, recovery, and distraction from their illnesses [11].

The effectiveness of aromatherapy against nausea and vomiting in the postoperative setting was evaluated by a Cochrane review [13]. In addition to available treatments against nausea and vomiting, aromatherapy was suggested as an additional option. Five randomized control trials (RCTs) and two controlled clinical trials (CCTs) included a total of 663 patients in this Cochrane review. This updated review was combined with nine previously included studies (six RCTs and three CCTs, with a total of 373 participants). The updated review included 16 studies including 1036 participants and concluded that, regarding the severity of nausea at the end of the treatment, aromatherapy may be as effective as the placebo. The evidence for these findings was low. In this review, the authors found that participants who received aromatherapy needed fewer antiemetic medications, but the evidence was also low-quality and hence uncertain. Patients receiving either aromatherapy or antiemetic medications seemed to report similar levels of satisfaction with their treatment, but again, evidence was based on low quality.

As aromatic oils are regularly used at the University Hospital Krems (UHK)’s palliative care ward, the aim of our study was to conduct a retrospective analysis of data on the effectiveness of using lemon oil pads as adjuvants to alleviate nausea and vomiting in patients with advanced cancers.

## 2. Materials and Methods

### 2.1. Background

At the UHK, various aroma care standards that apply to day-to-day care onwards are routinely implemented. Aromatherapy can be used in the following settings: room scenting, skin care of intact skin, applications for prophylaxis (e.g., pneumonia prophylaxis and decubitus prophylaxis), care-indicated washes, baths, partial baths, oral care when the oral mucosa is intact and rubs, and dry inhalation from a cotton pad. Aromatherapy is used very frequently in our palliative care ward. Before it is offered to patients, these patients’ medical histories are taken by the nurses. In addition to the usual questions, such as drug therapy, mental state, allergies, and intolerance, questions concerning scent preferences are mandatory to ask from patients. According to the standards of our hospital, educating patients about the use of aromatic oils in clinical settings and obtaining their consent before treatment starts are compulsory. Whether this is necessary in the form of a written declaration of consent is the responsibility of the respective facility. Consent is discussed verbally with the patient, but in the event of rejection or intolerance, this must be documented in the patient’s file.

### 2.2. Intervention

After the patients provided their consent, in case of nausea and vomiting, two drops of lemon oil were applied on commercially available cotton pads and then either clipped on the patients’ clothes (mostly on the collar) or given to the patient to use as needed. The chemical composition of the lemon oil used in our hospital was as follows: β-Pinen (15.0%), sabinen (2.0%), limonen (65.5%), y-terpinen (8.1%), β-caryophyllen (0.4%), neral (1.1%), alpha-terpinol (0.1%), nerylacetat (0.3%), geranial (1.6%), geranylacetate (0.3%), and evaporation residue (2.7%). The efficacy of the lemon pad application was evaluated after 10 min (yes/no). The patients were informed beforehand that regular rescue medication could be given at any time. The rescue medications provided in our palliative care unit included oral, sublingual, and intravenous HT-3 antagonists; oral metoclopramide; oral and intravenous metoclopramide; oral haloperidol; oral and intravenous diphenhydramine; oral and sublingual lorazepam; oral and intravenous dexamethasone; and oral or sublingual olanzapine.

Suspected causes for nausea and vomiting were categorized as suggested by Harris [14] and Leach [15] as follows: chemical (drugs and biochemical disturbance), impaired gastric emptying (gastric stasis and drugs), visceral/serosal (bowel obstruction and constipation), cranial (space occupying lesion and radiotherapy), vestibular (base of skull tumor), and cortical (anxiety and pain).

### 2.3. Aim of the Study

A retrospective evaluation of the medical records of all patients who received lemon pads against nausea and vomiting was conducted to evaluate whether there were improvements in these symptoms. We also evaluated the use of rescue medication. The efficacy of treatment with both lemon pads and rescue medication was recorded as yes/no.

### 2.4. Patients/Applications

All adult patients gave their consent to aroma care treatment against nausea and vomiting. All applications of lemon pads against nausea and vomiting were documented.

### 2.5. Data Collection

Data collection was performed using the medical records of patients who, with their consent, underwent lemon pad applications to relieve nausea and vomiting.

At the UHK, patient data are recorded in an access-restricted system. Access to patient data is recorded at any time via access authorization (personalized). The patient data relevant to the study were compiled and evaluated anonymously within the IT system of the UHK. Only authorized persons had access to the original data. The patients participating in the study were assigned a serial number ---). The study-related data were evaluated using this pseudonym number. This retrospective study involving human participants was conducted in accordance with the ethical standards of the institutional and national research committee and the 1964 Declaration of Helsinki and its later amendments or comparable ethical standards. The retrospective study was approved by the local ethics committee (GS4-EK-4/388-2016).

### 2.6. Statistics

Descriptive statistics were used to describe the data. The number of cases resulted from the number of applications. Differences between groups were calculated using chi-squared tests.

## 3. Results

### 3.1. Patient Characteristics and Number of Applications

A total of 66 patients received 222 applications of lemon oil pads against nausea and vomiting and were included in the analysis (Figure 1).

The median age at diagnosis was 68 years (range: 42–101; standard deviation: 12.1). Thirty-two (48.5%) patients were female, and 34 (51.5%) were male. All patients suffered from advanced haemato-oncological disease and were in a palliative setting in the palliative care ward. No further systemic anticancer treatment was administered to these patients (Table 1).

There were 17 patients with breast cancer (25.8%), 15 with lung cancer (22.7%), 13 with colorectal cancer (19.7%), 9 with ovarian cancer (13.6%), 5 with head and neck cancer (7.6%), 4 with gastric cancer (6.1%), 2 with renal cancer (3%), and 1 patient with myelodysplastic syndrome (1.5%) (Table 1).

A total of 222 applications of the lemon oil pads for nausea and vomiting were performed. Efficacy was measured as yes/no. No data were available for 17 applications. Adequate relief of nausea and vomiting was reported for 149 (73%) applications. Causes for nausea and vomiting were chemical for seven applications (3.2%), impaired gastric emptying for 44 applications (19.8%), visceral/serosal for 49 applications (22.1%), cranial for 34 applications (15.3%), vestibular for 14 applications (6.3%), and cortical for 74 applications (33.3%) (Table 1). No symptom control was seen for 56 (27%) applications (Figure 1 and Table 1). For the 56 applications without symptom control, first- and second-line rescue medications was successful in 53 and 3 cases, respectively.

After 53 (27%) applications to the patients, first-line rescue medication against nausea and vomiting was needed. The distribution of first-line rescue medication was as follows: 40 (20%) received sublingual ondasetron; 7 (4%) received intravenous metoclopramide; 2 (1%) received oral granisetron; 2 (1%) received intravenous diphenhydramine; and 2 (1%) received sublingual lorazepam. In three cases, second-line rescue medication was required: 2 (1%) received intravenous ondasetron, and one (0.5%) had intravenous dexamethasone (Table 1). Rescue medication as related to causes for nausea and vomiting was distributed as follows: chemical for 1 application of granisetron and 1 application of ondasetron; impaired gastric emptying for 5 applications of ondasetron and 7 applications of metoclopramide; visceral/serosal for 26 applications of ondasetron, 1 application of granisetron, and 1 application of diphenhydramine; cranial for 2 applications of ondasetron; vestibular for 2 applications of ondasetron; and cortical for 4 applications of ondasetron, 1 application of diphenhydramine, and 2 applications of lorazepam.

No third-line rescue medication for nausea and vomiting was necessary.

Regarding the number of applications of lemon oil pads per patient, 25 (37%) received one application only; 13 (20%) received 2 applications; 8 (12%) received 3 applications; 6 (9%) received 5 applications; 3 (4.5%) received 6 applications; 3 (4.5%) received 4 applications; and 2 (2%) received 9 applications. One patient each (10%) received 7, 8, 11, 12, and 15 applications.

### 3.2. Association between the Application of Lemon Oil Pads and Their Efficacy against Nausea and Vomiting

We evaluated whether there was an association between the application of lemon oil pads and their efficacy against nausea and vomiting after 10 min (yes/no). We found a statistically significant higher number of applications with efficacy versus no efficacy (*p* < 0.001). The application of lemon pads was 2.7 times likelier to be effective against nausea and vomiting. Of the 205 applications in 66 patients (with an average of 3 applications per patient), 149 (73%) were effective against nausea and vomiting, whereas 56 (27%) were ineffective.

When we evaluated the efficacy of lemon oil pads regarding the cause of nausea and vomiting, we found that efficacy was most frequent in nausea and vomiting related to cortical issues (anxiety and pain) (59 applications (39.6%)) (Table 2). When we grouped cortical causes versus all the other causes, no efficacy was more prominent in all the other causes than in cortical causes. When we grouped impaired gastric emptying/visceral serosal versus all the others, efficacy was significantly higher in all the other causes as compared to in impaired gastric emptying/visceral serosal (Table 2).

## 4. Discussion

Antiemetics have side effects, such as QT (QT interval is the time from the start of the Q wave to the end of the T wave, time taken for ventricular depolarisation and repolarisation) prolongations (e.g., 5-HT3 receptor antagonists), cardiac dysrhythmias (e.g., droperidol), drowsiness (e.g., antihistamines), and sleep disorders (e.g., dexamethasone). A reduction in the need for antiemetics by using aromatherapy against nausea and vomiting could lead to fewer side effects for this vulnerable group of patients.

Lemon oil can reduce sluggishness, stimulate the central nervous system and serve as stimulants to the central nervous system [16]. Aromatherapy is often associated with the sense of smell. Inhalation is one method to unfold the efficacy of essential oils. By smelling an appropriate essential oil, the patient can become relaxed or invigorated, be soothed or even can have the sensation of reduced pain, nausea, anxiety, or even dyspnea. Several studies have used it against nausea and vomiting. Naturally pure essential lemon oil is obtained by pressing the fruit peel, since the essential oil is stored in small oil containers in the peel of the citrus fruit. Like all citrus oils, also called citrus oils, lemon oil is also rich in monoterpenes, especially limonene [16].

The use of lemon oil pads as aromatherapy in palliative care against nausea and vomiting is very limited. We found only one study in the literature describing the use of lemon aroma sticks in patients in a retrospective audit [17]. The authors saw a benefit of lemon aroma sticks in reducing nausea and vomiting in cancer patients [17]. Reis et al. described the use of orange aromatherapy against nausea and vomiting [18]. To our knowledge, no studies investigated the use of lemon oil on cotton pads against nausea and vomiting in patients with advanced cancer. We chose lemon oil as aromatherapy, because there are some studies that describe a positive effect of lemon oil against nausea and vomiting in pregnancy [19,20].

In our retrospective analysis, we found that the use of lemon oil pads was effective in 73% of all applications and that rescue medication was necessary in only 27% of all applications. When we grouped impaired gastric emptying/visceral serosal versus all the others, efficacy was significantly higher in all the other causes as compared to in impaired gastric emptying/visceral serosal. This indicated that aromatherapy against nausea and vomiting is not effective in mechanical causes leading to these symptoms. These patients should not be offered aromatherapy firsthand. Although it was a retrospective study, our results showed a benefit for the use of lemon pads against nausea and vomiting in selected causes. When we evaluated which patients needed rescue medication, we found that in cases where visceral or serosal problems were present, lemon pads were not efficient in 27 patients, but 5-HT-3 antagonists. This can be explained by the fact that in these patients, due to the irritation of the bowel wall, 5-HT-3 was released, inducing nausea and vomiting. Hence, the use of lemon oil pad should be avoided in such causes because the use of 5-HT-3-antagonists is undisputed in this indication [14]. As for impaired gastric emptying, metoclopramide was successful as rescue medication in seven patients. This is also explained by the literature, where metoclopramide is standard in conditions with impaired gastric emptying without complete bowel obstruction [14].

The literature on the effectiveness of aromatherapy compared with the placebo in improving the quality of life and alleviating symptoms, such as nausea, anxiety, sleep disturbances, and pain, in patients with advanced cancers in palliative care units is sparse [21,22,23,24,25,26,27].

Regarding aromatherapy against nausea and vomiting, there are some studies describing an effect in patient collectives other than palliative care. Two RCTs examined the effectiveness of aromatic oil in chemotherapy- and radiotherapy-induced nausea [24,25]. Lua et al. tested the use of ginger oil in patients with breast cancer [17]. The inhalation of ginger essential oil significantly reduced the VAS (visual analogue scale) nausea score as compared to placebo in this study. Nevertheless, no significant effect of ginger oil inhalation was seen for vomiting.

The study by Tayarani tested the essential oils Mentha spicata (spearmint) and Mentha × piperata (peppermint) against chemotherapy-induced nausea and vomiting (CINV), showing a significant reduction in the intensity and frequency of nausea [25].

Few studies have investigated the use of aromatherapy and nausea and vomiting in a palliative care setting [7,21,28]. The benefits of aromatherapy were reported in a recent study, focusing on symptoms frequent in palliative care patients such as pain, nausea, vomiting, anxiety, sadness, stress, sleeplessness, and agitation [29].

To date, there has been more literature describing the efficacy of aromatherapy against nausea and vomiting in other settings such as postoperative nausea and vomiting (PONV) and CINV than in nausea and vomiting in palliative care patients. The effects of aromatherapy in patients with PONV in emergency departments (EDs) or in patients with CINV were investigated in a couple of randomized studies. Treatment for PONV and CINV include 5-HT3 receptor antagonists, antihistamines, or D-2 receptor antagonists. An effective complementary treatment for nausea, both postoperatively and in other settings, seems to be aromatherapy with alcohol or peppermint [30,31].

In a randomized, double-blind, placebo-controlled study, 33 patients who developed postoperative nausea were randomized to aromatherapy with either isopropyl alcohol peppermint oil or saline (placebo) [32]. In this study, not only aromatherapy, but also the saline placebo, reduced the perceived severity of postoperative nausea, suggesting that the beneficial effect was related rather to controlled breathing patterns than to aromatherapy inhaled [32].

It can be difficult to treat nausea and vomiting in EDs. Neither intravenous metoclopramide nor ondasetron was superior to the placebo against nausea in an ED setting. Nevertheless, metoclopramide reduced the need for rescue antiemetics [33].

One hundred twenty-two adults in an ED suffering from mild to moderate nausea, mostly from infectious gastroenteritis, were randomized to receive inhaled isopropyl alcohol, ondansetron, or both [34]. A significant reduction in nausea was seen after 30 min.

For inhaled isopropyl alcohol, four RCTs including a total of 215 patients were evaluated in a systematic review, indicating that fewer patients needed rescue antiemetics versus standard therapy (26% vs. 39% (placebo)) [13]. In another study, 84 patients were randomized to receive either inhaled isopropyl alcohol or saline-soaked pads for the placebo, showing less nausea in the isopropyl alcohol group [35].

One of the most common and feared side effects of anticancer treatment is CINV. CINV leads to a decreased quality of life for cancer patients and is associated with a higher rate of morbidity. The potential role of aromatherapy against CINV was investigated in several studies. The efficacy of inhaled aromatherapy against CINV in addition to standard treatment was investigated in a systematic review including 11 studies [36].

Three of the seven studies found that nausea is decreased by using chamomilla, ginger, or cardamon essential oil [36]. Altogether, this review suggested that the use of direct inhaled aromatherapy is beneficial in managing CINV. Nevertheless, the authors concluded that larger studies are needed to prove the usefulness of inhaled aromatherapy against CINV [36].

The effects of using a cool damp washcloth with peppermint essential oil versus using a cool damp washcloth alone to reduce nausea were tested in 79 patients receiving chemotherapy [37]. The use of peppermint oil was more effective in reducing the intensity of nausea as reported by the patients when compared to the use of a cool washcloth alone [37].

Peppermint oil was also used in a quasi-randomized controlled study. Patients received one drop of peppermint oil on their upper lips three times a day for five days after chemotherapy in addition to routine antiemetic treatment or only routine antiemetic treatment. The frequency of nausea, vomiting, and retching was significantly reduced in the peppermint oil group [38].

The findings that aromatherapy is useful in patients with PONV, in emergency room settings, and in patients with CINV could enforce research regarding aromatherapy for patients with advanced cancers in a palliative care setting.

The studies investigating the role of the efficacy of aromatherapy against CINV and PONV could serve as templates for designing prospective studies using aromatherapy, especially lemon oil pads against nausea and vomiting in patients in a palliative setting.

Our study had several major limitations. The weaknesses of our study were that it was a retrospective analysis. The retrospective design made the analysis of the outcome parameters very difficult. Another major limitation of our study was that no control group without lemon pads was used to confirm our hypothesis. We had no control group evaluating a placebo effect. As nausea and vomiting may be resolved spontaneously, the conclusions of the effectiveness of aromatherapy without a control arm were very limited.

Other major limitations of our study were that we probably also observed a strong placebo effect for the application of lemon oil pads, which can only be ruled out by providing a control arm using a placebo as a comparator. A placebo effect can be observed in studies which track subjective outcomes on a continuous scale [39,40] or as the case in our retrospective analysis, by evaluating whether the intervention was effective as documented by yes/no.

Regarding the obsequiousness bias, we believe that patients in our retrospective analysis did not alter their subjective responses about the efficacy of the lemon pad against nausea and vomiting, because the sensation of these symptoms are very tantalizing [41].

Therefore, we concluded that in a prospective trial, the confounder obsequiousness bias should be calculated but is not expected to have a major impact on the outcome of the results, whereas the implementation of a control arm with a placebo is mandatory.

As there are already numerous randomized and unrandomized studies about aromatherapy in palliative care in the literature, the results of our study will not lead to establishing aromatherapy in treatment of nausea and vomiting in patients with advanced cancer.

## 5. Conclusions

In our retrospective analysis, we found that the application of lemon oil pads was effective in reducing nausea and vomiting in patients with advanced cancers in a palliative care setting.

Although meta-analyses and studies did not find any significant benefits of aromatherapy for palliative care patients, especially those who experience nausea and vomiting, it is well accepted by patients. It also seemed to have positive effects against nausea and vomiting in our patients. For severe symptoms, such as nausea and vomiting in palliative care patients, rescue medication will always remain compulsory.

As there are no large RCTs investigating the effects of aromatherapy against nausea and vomiting in palliative care patients, such studies should be performed in the future. Randomized trials investigating the efficacy of aromatherapy in patients with PONV, in emergency room settings, or in patients with CINV could serve as models for similar trials in patients with advanced cancers in a palliative care setting. These could help further evaluate the role of aromatherapy against nausea and vomiting in this patient population.

## Figures and Tables

**Figure 1 cancers-14-02131-f001:**
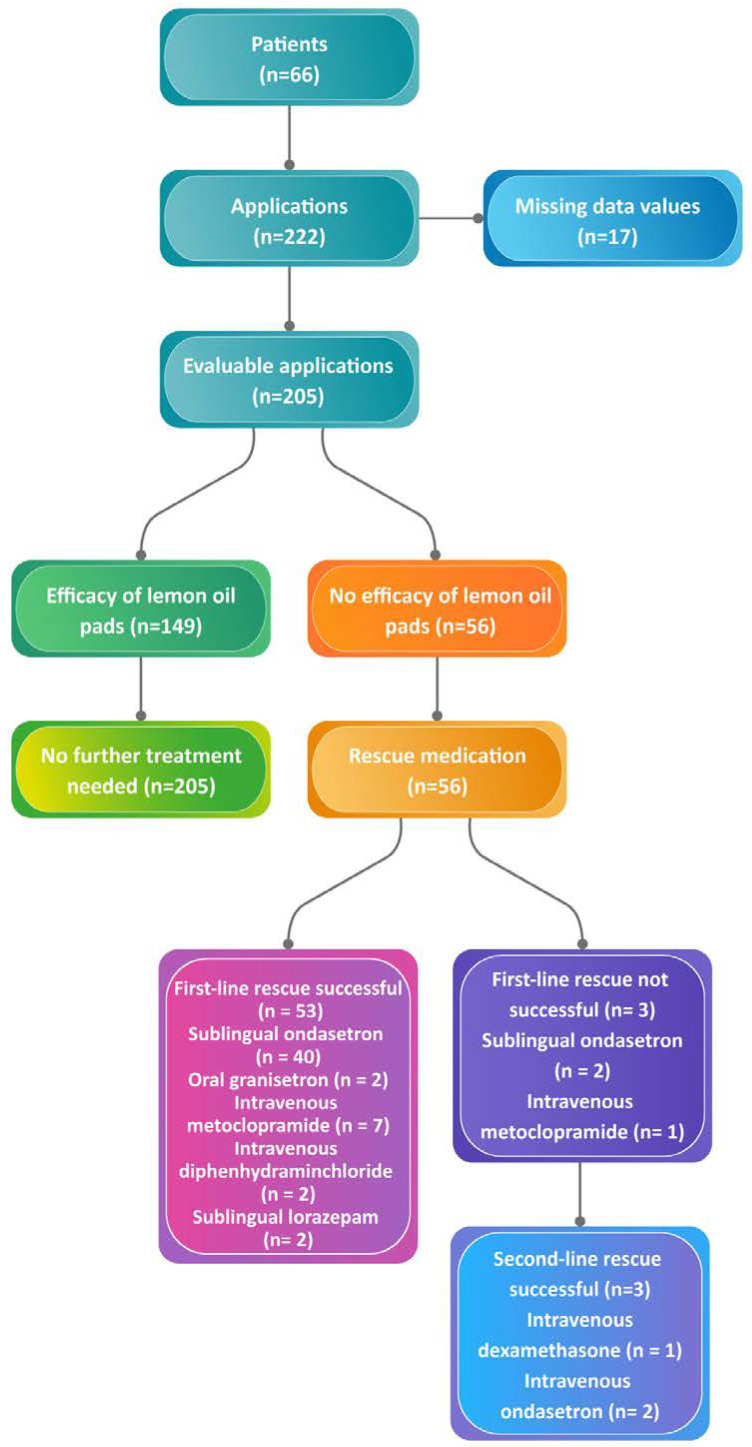
Efficacy of aromatherapy applications (lemon oil pads) against nausea/vomiting. Sixty-six patients received a total of 222 aromatherapy applications. Data were missing from 17 applications. Two hundred five applications were evaluated. Lemon oil pads were efficient in 149 applications, and no further rescue medication was needed. No efficacy of lemon oil pads was seen for 56 applications. First-line rescue medication was successful in 53 applications. Second-line rescue medication was successful in the remaining 3 applications.

**Table 1 cancers-14-02131-t001:** Patients’ (*n* = 66) characteristics and descriptions of lemon pad applications (*n* = 222).

All Patients		66
Gender	Female	32 (48.5%)
	Male	34 (51.5%)
Age	Median (range), years	68 (42–101)
Advanced disease	Breast cancer	17 (25.8%)
	Lung cancer	15 (22.7%)
	Colorectal cancer	13 (19.7%)
	Ovarian cancer	9 (13.6%)
	Head and neck cancer	5 (7.6%)
	Gastric cancer	4 (6.1%)
	Renal cancer	2 (3%)
	Myelodysplastic syndrome	1 (1.5%)
Causes for nausea and vomiting	Chemical	7 (3.2%)
	Impaired gastric emptying	44 (19.8%)
	Visceral/serosal	49 (22.1%)
	Cranial	34 (15.3%)
	Vestibular	14 (6.3%)
	Cortical	74 (33.3%)
All applications of lemon pads		222
	Efficacy data available	205
	Efficacy data not available	17
Indication for the application of lemon pads		222
	Nausea only	210 (94.6%)
	Nausea and vomiting	12 (5.4%)
First-line rescue medication against nausea and vomiting		53 (27%)
	Sublingual ondasetron	40 (20%)
	Intravenous metoclopramide	7 (4%)
	Oral granisetron	2 (1%)
	Intravenous diphenhydramine	2 (1%)
	Sublingual lorazepam	2 (1%)
Second-line rescue medication against nausea and vomiting		3 (2%)
	Intravenous ondasetron	2 (1%)
	Intravenous dexamethasone	1 (0.5%)

**Table 2 cancers-14-02131-t002:** Cross-tabulation: efficacy of lemon pad applications related to causes for nausea and vomiting (*n* = 205).

Efficacy of Lemon Pads	Causes	Frequency
No	Cortical All others	9 (4.4%) 47 (22.9%)
Yes	Cortical All others	59 (39.6%) 90 (43.9%)
No	Impaired gastric emptying/visceral serosal All the others	40 (47.1%) 16 (13.3%)
Yes	Impaired gastric emptying/visceral serosal All the others	45 (30.2%) 104 (69.8%)

## Data Availability

The data presented in this study are available on request from the corresponding author. The data are not publicly available.

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
