# Peer review of "Aromatherapy in Palliative Care: A Single-Institute Retrospective Analysis Evaluating the Effect of Lemon Oil Pads against Nausea and Vomiting in Advanced Cancer Patients"

_cancers, 2022, doi:10.3390/cancers14092131_

Round 1
Reviewer 1 Report
This paper "Aromatherapy in palliative care: A single-institute retrospective analysis evaluating the effect of lemon oil pads against nausea and vomiting in advanced cancer patients" is very interesting and original. I think that it is line with the readership of Cancers, but some points need to be revised before publication:
-the table 1 is not clear, please add columns and lines.
-please add also the chemical composition of lemon oil pads.
Author Response
Response to Reviewer 1 comments
Dear Reviewer,
Thank you very much for your time to review our manuscript and adding all those valuable comments! We tried to modify our manuscript according to your suggestions as follows:
Point 1. “-the table 1 is not clear, please add columns and lines.”:
Response 1. We changed table 1 and table 2 to a proper table each with columns and lines.
Point 2. “-please also add the chemical composition of lemon oil pads.”
Response 2. We added the chemical composition of the lemon oil applied to the commercially available cotton pads in part 2.2 (Intervention).
If you have further questions, please do not hesitate to contact us!
Kind regards,
Gudrun Kreye
Reviewer 2 Report
This manuscript is an original article that retrospectively evaluated the efficacy of lemon oil pads against nausea and vomiting in advanced cancer patients.
The authors showed that lemon oil pads were effective in 149 of 222 (73%) applications without any side effects.
This study was conducted well, and the results will be of interest to clinicians in the field.
However, the following major and minor issues require clarification:
Major
- The description in the Introduction, Discussion and Conclusion is redundant. It should be summarized focused on aromatherapy using lemon oil pads in palliative care against nausea and vomiting.
- The authors should introduce some information regarding mechanism and utility of lemon oil pads as an aromatherapy.
- Please provide the data regarding the presumed causes of nausea and vomiting. Furthermore, I recommend the authors discuss what type of nausea lemon oil was effective or not for based on the presumed causes and the rescue doses.
Minor
- (Conclusion) The limitation should be written in the Discussion section in more detail. The description regarding side effects should be written in other section.
- Table 2 has little information for occupied space. Please modify it.
Author Response
Response to Reviewer 2 comments
Dear Reviewer,
Thank you very much for your time to review our manuscript and adding all those valuable comments! We tried to modify our manuscript according to your suggestions as follows:
Major
- Point 1. The description in the Introduction, Discussion and Conclusion is redundant. It should be summarized focused on aromatherapy using lemon oil pads in palliative care against nausea and vomiting.
Response 1. We condensed the introduction by summarizing studies related to complementary and integrative medicine in general and tried to focus on aromatherapy against nausea and vomiting. There are very few clinical studies dealing with the application of lemon oil pads for patients with advanced cancer. There is one study using lemon aroma sticks to reduce nausea and vomiting in patients in advanced cancer (Dyer et al.). Another study used another citrus fruit, not lemon, but orange (Reis et al.). Additionally, we described other studies evaluating the efficacy of lemon oils in other conditions such as nausea and vomiting in pregnancy (Kustrivanti, Safajou).
- Point 2. The authors should introduce some information regarding mechanism and utility of lemon oil pads as an aromatherapy.
Response 2. We added some information regarding the mechanism and utility of lemon oil as aromatherapy in the introduction.
- Point 3. Please provide the data regarding the presumed causes of nausea and vomiting. Furthermore, I recommend the authors discuss what type of nausea lemon oil was effective or not for based on the presumed causes and the rescue doses.
Response 3. We provided data regarding the presumed causes of nausea and vomiting in the material and method section, 2.2. intervention. As recommended by Harris and Leach, we grouped causes for nausea and vomiting in patients with advanced cancer as follows: chemical (drugs, biochemical disturbance), Impaired gastric emptying (gastric stasis, drugs), visceral/serosal (bowel obstruction, constipation), cranial (space occupying lesion, radiotherapy) , vestibular (base of skull tumor), cortical (anxiety, pain).
In the result section, we described the results for different causes for nausea and vomiting in our patient group. We found that lemon oil pads were less effective in patients who had impaired gastric emptying, bowel obstruction or constipation. We show our results also in table 1 and table 2.
Minor
- Point 1. (Conclusion) The limitation should be written in the Discussion section in more detail. The description regarding side effects should be written in other section.
Response 1. We described the limitations of our study more in detail and moved it from the conclusion section to the discussion section.
- Point 2. Table 2 has little information for occupied space. Please modify it.
Response 2. We deleted table 2 because it has little information which is already sufficiently described in the text.
If you have further questions, please do not hesitate to contact us!
Kind regards,
Gudrun Kreye
Reviewer 3 Report
The authors conducted a retrospective analysis on aromatherapy for nausea and vomiting in the palliative ward. Because nausea and vomiting may resolve spontaneously or by the placebo effect, it is impossible to make sure the effectiveness of aromatherapy without a control arm. The retrospective design makes it even more difficult to confirm the validity of the outcome parameters. There are already numerous flawed studies about aromatherapy in the literature. I don't think this article can bring anything new to this field. Meanwhile, figure 1 has several typos.
Author Response
Response to Reviewer 3 comments
Dear Reviewer,
Thank you very much for taking your time to review our manuscript.
Please find our responses as follows:
Comments and Suggestions for Authors
Point 1. Figure 1 has several typos.
We corrected the typos in figure 1.
Kind regards, Gudrun Kreye
Round 2
Reviewer 2 Report
The revised manuscript is improved. However, the following issues should be addressed.
Major
- The description in the Discussion is still redundant. It should be summarized focused on aromatherapy using lemon oil pads in palliative care against nausea and vomiting based on the results in the present study.
- The results regarding the causes of nausea are interesting. However, the statistical analyses and discussion based on the analyzed results are insufficient. Furthermore, I recommend the authors discuss what type of nausea lemon oil was effective or not for based on the rescue doses.
Minor
- (Conclusion) The description regarding side effects should be written in other section.
- Table 3 seems somewhat confusing. Please modify it as readers easily understand.
Author Response
Dear Reviewer,
Thank you very much for taking again your time to review our manuscript and adding all those valuable comments! We tried to modify our manuscript according to your suggestions as follows:
Major
Point 1. "The description in the Discussion is still redundant. It should be summarized focused on aromatherapy using lemon oil pads in palliative care against nausea and vomiting based on the results in the present study."
In the discussion, we tried to focus on aromatherapy using lemon oil pads in palliative care against nausea and vomiting based on the results in the present study.
Because there are almost no studies on aromatherapy against nausea and vomiting in palliative care, especially studies with lemon oil pads, we mentioned some studies using other oils in other patient collectives such as patients who suffer from postoperative nausea and vomiting, patients receiving chemotherapy, pregnant women, and patients in emergency department settings. We referred to this literature because they describe the use of aromatherapy against nausea and vomiting. Therefore, we discussed the studies evaluating the efficacy of aromatherapies other than lemon oil pads for palliative care patients in our discussion. As they could also serve as role models for studies investigating the use of aromatherapy in palliative care settings, we discussed them more detailed.
Point 2. "The results regarding the causes of nausea are interesting. However, the statistical analyses and discussion based on the analyzed results are insufficient. Furthermore, I recommend the authors discuss what type of nausea lemon oil was effective or not for based on the rescue doses*."
Based on the present data, more complex statistical analyses are not possible to our opinion. Please provide further information which further statistical analyses should be performed.
*We tried to discuss more detailed in the discussion what type of nausea lemon oil was effective or not for based on the rescue doses. Beforehand, we included results of what type of nausea needed rescue medication in the result part.
Minor
Point 1. "(Conclusion) The description regarding side effects should be written in other section."
We moved the description regarding side effects from the conclusion to the beginning of the discussion.
Point 2. "Table 3 seems somewhat confusing. Please modify it as readers easily understand."
We deleted table 3 completely already in the first run and included the results in the text as there was only one result present in table 3 and easily transferable to the text. If you are referring to table 2, we modified it and reduced the content to the grouped causes for nausea and vomiting.
If you have further questions, please do not hesitate to contact us!
Kind regards, Gudrun Kreye
Reviewer 3 Report
The authors failed to correct the fundamental design issue I raised in my review. I still do not believe the conclusion of the study can be supported by the current data.
Author Response
Dear Reviewer,
Thank you very much for taking again your time to review our manuscript. Our conclusions of the present study are based on retrospective data, therefore we presented them in the conclusion, even though results of a retrospective study cannot be as valuable as prospective data. We regret that we only have retrospective data now. Nevertheless, the results we found in our retrospective study showed a benefit for the use of lemon oil pads in our institute. We are planning to use the results of this study to plan a prospective study.
If you have further questions, please do not hesitate to contact us!
Kind regards, Gudrun Kreye